# Patterns of Hearing Loss in Irradiated Survivors of Head and Neck Rhabdomyosarcoma

**DOI:** 10.3390/cancers14235749

**Published:** 2022-11-23

**Authors:** Franciscus A. Diepstraten, Jan Wiersma, Reineke A. Schoot, Rutger R. G. Knops, Charlotte L. Zuur, Annelot J. M. Meijer, Raquel Dávila Fajardo, Bradley R. Pieters, Brian V. Balgobind, Henrike Westerveld, Nicole Freling, Harm van Tinteren, Ludwig E. Smeele, Arjan Bel, Marry M. van den Heuvel-Eibrink, Robert J. Stokroos, Johannes H. M. Merks, Alexander E. Hoetink, Marinka L. F. Hol

**Affiliations:** 1Princess Máxima Center for Pediatric Oncology, 3584 CS Utrecht, The Netherlands; 2Department of Radiation Oncology, Amsterdam University Medical Centers, University of Amsterdam, 1105 AZ Amsterdam, The Netherlands; 3Department of Head and Neck Surgery, Netherlands Cancer Institute, 1066 CX Amsterdam, The Netherlands; 4Department of Otorhinolaryngology Head and Neck Surgery, Leiden University Medical Center, 2333 ZA Leiden, The Netherlands; 5Department of Radiation Oncology, University Medical Center Utrecht, 3584 CX Utrecht, The Netherlands; 6Amsterdam Cancer Center, Cancer Treatment and Quality of Life, 1081 HV Amsterdam, The Netherlands; 7Department of Radiology, Amsterdam University Medical Centers, University of Amsterdam, 1105 AZ Amsterdam, The Netherlands; 8Trial and Data Center, Princess Máxima Center for Pediatric Oncology, 3584 CS Utrecht, The Netherlands; 9Department of Maxillofacial Surgery, Amsterdam University Medical Centers, 1105 AZ Amsterdam, The Netherlands; 10Department of Otorhinolaryngology-Head and Neck Surgery, University Medical Center, 3584 CX Utrecht, The Netherlands; 11University Medical Center Utrecht Brain Center, 3584 CX Utrecht, The Netherlands

**Keywords:** head and neck rhabdomyosarcoma, radiotherapy, ototoxicity, audiological monitoring, survivorship

## Abstract

**Simple Summary:**

Hearing loss (HL) can be a side effect of paediatric cancer treatment and can be caused by chemotherapy but also local therapies such as radiotherapy and/or surgery of the head and neck region. In this study, the frequency and patterns of HL were assessed in survivors of head and neck rhabdomyosarcoma (HNRMS). Our secondary aim was to look into the dose–effect relationship between radiotherapy dose on the cochlea and the presence of HL. Forty-nine survivors of HNRMS were included in this study, forty-two of them underwent audiological evaluation. HL was found in up to 19% of the survivors. Four survivors had low frequencies HL with normal hearing or milder HL in the higher frequencies. In our series, HL (≥Muenster 2b) was significantly associated with the maximum cochlear irradiation dose (*p* = 0.047). More research is needed on HL patterns in HNRMS survivors and on the radiotherapy dose–effect relationship.

**Abstract:**

Purpose: The frequency and patterns of HL in a HNRMS survivor cohort were investigated. A dose–effect relationship between the dose to the cochlea and HL was explored. Methods: Dutch survivors treated for HNRMS between 1993 and 2017 with no relapse and at least two years after the end of treatment were eligible for inclusion. The survivors were evaluated for HL with pure-tone audiometry. HL was graded according to the Muenster, Common Terminology Criteria for Adverse Events (CTCAE) v4.03 and International Society for Paediatric Oncology (SIOP) classification. We defined deleterious HL as Muenster ≥ 2b, CTCAE ≥ 2, and SIOP ≥ 2. Mixed-effects logistic regression was used to search for the dose–effect relationship between the irradiation dose to the cochlea and the occurrence of HL. Results: Forty-two HNRMS survivors underwent pure-tone audiometry. The Muenster, CTCAE and SIOP classification showed that 19.0% (*n* = 8), 14.2% (*n* = 6) and 11.9% (*n* = 5) of survivors suffered from HL, respectively. A low-frequency HL pattern with normal hearing or milder hearing loss in the higher frequencies was seen in four survivors. The maximum cochlear irradiation dose was significantly associated with HL (≥Muenster 2b) (*p* = 0.047). In our series, HL (≥Muenster 2b) was especially observed when the maximum dose to the cochlea exceeded 19 Gy. Conclusion: HL occurred in up to 19% of survivors of HNRMS. More research is needed on HL patterns in HNRMS survivors and on radiotherapy dose–effect relationships.

## 1. Introduction

Rhabdomyosarcoma (RMS) is a primitive malignant soft tissue sarcoma of the skeletal muscle phenotype that originates from mesenchymal cells [1]. It is the most common childhood and adolescent soft tissue sarcoma (3–5% of childhood tumours, 50% of paediatric soft tissue sarcomas) [1,2,3,4]. The median age at diagnosis of RMS patients is 6.5 years, and there is a slight male predominance (male/female ratio 1.3/1) [5]. Different subtypes exist, including embryonal (70%), alveolar (20%), and spindle cell/sclerosing (10%) RMS [1,2,3,4,5]. Local treatment options for RMS include external beam radiotherapy with photons (RT), external beam radiotherapy with protons (PT) [6], Ablative surgery followed by the MOuld technique after loading brachytherapy and surgical REconstruction (AMORE) [7], and the combination of RT or PT with conventional surgery [8]. With improvements in multimodality treatment, imaging, and supportive care, five-year overall survival has increased up to 70–90%, depending on the patient and tumour characteristics [9]. As survival rates increase, more attention is needed to inventory and prevent adverse effects of treatment such as dental and facial deformities [10,11,12], endocrine disorders [13], and hearing loss (HL) [14].

Due to the complex anatomy of the head and neck region and the extension of the tumour, radiotherapy and/or surgery can directly or indirectly affect the nasopharynx, middle ear, nerves, brain, and cochlea and consequently result in conductive, sensorineural, or mixed HL [14,15,16,17]. In addition, recurrent ear nose throat (ENT) infections might contribute to reversible HL in HNRMS patients [14]. It seems that radiotherapy has a dose-dependent effect on hearing outcomes [18,19,20]. Where chemotherapy mainly induces irreversible hearing loss in an early stage during treatment [21], radiation-induced HL is mainly described as a late effect in paediatric cancer patients, especially in those who receive a high cochlear cumulative radiation dose (>30 Gy) [15,22,23]. Pre-treatment hearing levels appeared to be a predictive factor for hearing capability after chemotherapy and radiotherapy [24]. Younger patients are more likely to develop HL. This is important, as HL is a serious problem which has negative consequences for speech and language development in children [25]. This can result in reduced communication skills with consequences for psychosocial and socioeconomic development. A lower quality of life is reported by children with HL compared to peers with normal hearing [26].

As survivors with HNRMS are treated with three different radiotherapy modalities in our cohort, we can investigate hearing function in these groups. The primary aim of the current study is to describe the frequency and pattern of hearing loss as measured by pure-tone audiometry and as systematically assessed by frequently used ototoxicity grading scales in HNRMS survivors treated with either RT, PT, or brachytherapy (BT) (BT as part of AMORE). The secondary aim of this study was to explore possible dose–effect relationships between the radiation dose to the cochlea and the development of HL.

## 2. Materials and Methods

### 2.1. Survivors

Survivors were selected through two different identification systems. First, survivors of HNRMS, treated between January 1993 and December 2017 and visiting the follow-up clinic at the Emma Children’s Hospital at least two years after the completion of treatment, were selected for inclusion. Secondly, HNRMS survivors treated in one of the paediatric oncology centres in the Netherlands who visited the Princess Máxima Center late effects department for audiological follow-up between 2018 and March 2021 were eligible for inclusion. From 2018 onwards, all paediatric oncology care, including follow-up, was centralised in a national centre, the Princess Máxima Center for Pediatric Oncology. Inclusion criteria were: (1) treated for primary HNRMS disease, and (2) no recurrence of disease at least two years after the completion of treatment. Survivors underwent an audiological examination in the Emma Children’s Hospital, Amsterdam Medical Center (AMC) and/or Princess Máxima Center, University Medical Center Utrecht (UMCU), during a follow-up multidisciplinary outpatient clinic consultation.

Audiological data were collected during regular follow-up visits. The institutional Review Board of the Amsterdam University Medical Center decided that the Act on Medical Research Involving Subjects did not apply.

### 2.2. Treatment

The included HNRMS survivors had been treated according to SIOP–MMT (International Society for Paediatrics Oncology–Malignant Mesenchymal Tumour group) 95 [27], SIOP–MMT 89 [28,29], E*p*SSG (European *paediatric* Soft tissue sarcoma Study Group) RMS 2005 [30], CWS (Cooperative WeichteilSarkom Studien Gruppe)-91 [31], or CWS-2007HR [32] protocols. In SIOP–MMT 95 and SIOP–MMT 89, children received ifosfamide, vincristine, actinomycin D (IVA), or IVA alternating with carboplatin, epirubicin, and etoposide. The maximum cumulative carboplatin dose was 3600 mg/m^2^. Survivors treated according to E*p*SSG RMS 2005 protocol received IVA or IVA with doxorubicin; high-risk patients were randomised to receive maintenance chemotherapy with vinorelbine and cyclophosphamide. Survivors treated according to CWS-91 received etoposide, vincristine, dactinomycin, ifosfamide, and doxorubicin. Survivors treated according to CWS-2007 received IVA. All survivors had received local treatment with RT, PT, or BT (as part of AMORE).

### 2.3. Radiotherapy Data

The original radiotherapy treatment data were retrieved for all survivors treated at the Emma Children’s Hospital. The cochlea was delineated by one researcher (MH) and checked by a head and neck radiologist (NF). The cochlea was delineated on a computed tomography scan (bone setting) according to a head and neck anatomy app [33]. This head and neck anatomy app is a fully interactive atlas of head and neck anatomy created by doctors for anyone else professionally involved in head and neck anatomy [33]. Dose to organs was exported for minimal, mean, and maximum doses (D0.1 cm^3^). To enable comparisons between treatment modalities, all radiotherapy parameters were recalculated as equivalent doses in 2 Gy fractions (EQD2) using the following formula; EQD_2_ = BED/[2 + (α/β)]. The BED depends on radiotherapy modality (RT, PT, BT) and is described by the following generic formula: BED = D*q_let_*[g_repair_*d + (α/β)], with D representing total dose, d is the dose per fraction, g_let_ is a factor representing the effective dose relative to photon therapy, with g_let_ = 1 for RT and BT, and g_let_ =1.1 for PT, and g_repair_ is a factor describing the change in the dose–effect due to the biological response, i.e., “damage” repair during and in between pulsed (BT-PDR) or during (BT-LDR) dose delivery. The alpha–beta ratio, a parameter that can be assessed and which represents the measure of cell death and cell repair, for the cochlea was set at 2 Gy [34]. The parameter describing the influence of repair is the half-time for repair, and it was set to 1.5 h [35,36].

### 2.4. Audiometry

Hearing function was measured by pure-tone audiometry (PTA) during follow-up visits after HNRMS treatment. Air-conducted thresholds were measured between 0.125–8 kHz, and if elevated thresholds were found, bone-conducted thresholds were measured between 0.5–4 kHz. HL was evaluated by two researchers (M.L.F.H., F.A.D.) and an audiologist (A.E.H.). HL was graded according to the Muenster, International Society for Paediatric Oncology (SIOP) and U.S. National Cancer Institute Common Terminology Criteria for Adverse Events (CTCAE) v4.03 criteria (Appendix A) [37,38,39]. Disagreements between the researchers (M.L.F.H., F.A.D.) and the audiologist (A.E.H.) were discussed. The audiologist (A.E.H) made the final decision. Deleterious HL was defined as a Muenster ≥ 2b grade, SIOP ≥ 2 grade, or CTCAE grade ≥ 2. Both ears were graded independently.

### 2.5. Statistics

Descriptive statistics were used, including the median and range for the quantitative variables of age and follow-up time, and the frequency and percentage for the qualitative variables of sex, RMS subtype, and treatment strategy. The characteristics between the groups with and without hearing loss were compared using Pearson’s Chi-square test for categorical variables, an independent Student’s t-test for normally distributed continuous variables, or a Mann–Whitney U test for abnormally distributed continuous variables. The dose–effect evaluation for radiotherapy data on HL was analysed using mixed-effects logistic regression with the patient as the random factor. A two-sided *p*-value less than 0.05 was considered statically significant. Statistical analyses were performed using IBM SPSS Statistics 25.0.0.2 (SPSS Inc. Chicago, IL, USA) and R version 4.2.1 (2022-06-23 ucrt).

## 3. Results

### 3.1. Clinical Characteristics

Forty-nine survivors (25 females, 24 males) of primary HNRMS were included in this study (Table 1). Thirty-nine HNRMS survivors initially treated at the Emma Children’s Hospital were identified, of whom 32 underwent audiological testing during follow-up visits. Additionally, 10 HNRMS survivors with audiological tests during follow-up at the Princess Máxima Center were identified (Appendix A): six of these patients were treated at the Emma Children’s Hospital, three at the Sophia Children’s Hospital and one patient at the University Medical Center Groningen. The survivors had a median age of 5.0 years (range 0.04–13.4) at the time of diagnosis and a median age of 16.3 years (range 5.2–33.7) at the time of follow-up. The median follow-up time was 10.1 years (range 2.0–26.6). Forty-three survivors had been treated for embryonal RMS, five survivors had alveolar RMS, and in one survivor, the RMS histology subtype was unknown. In total, 51% (25/49) of the survivors had a parameningeal tumour location, 37% (18/49) had an orbital location, and 12% (6/49) had a non-parameningeal localisation. Twenty-eight survivors had been treated according to the EpSSG RMS 2005 protocol, thirteen according to the SIOP–MMT-95 protocol, five according to SIOP–MMT-89, two according to CWS-2007HR, and one to CWS-91. Seventeen survivors were treated with RT, twenty-three with BT (as part of AMORE), and nine with PT (Table 1 and Table 2).

### 3.2. Frequency and Patterns of Hearing Loss

Based on Muenster, CTCAE, and SIOP grading, 19.0% (8/42), 14.3% (6/42), and 11.9% (5/42) of the survivors had deleterious HL, respectively (Table 1). There were no statistically significant differences regarding clinical characteristics such as age at diagnosis (*p* = 0.863) and follow-up time (*p* = 0.235) between the survivors with HL and those without HL. A combination of conductive and sensorineural HL was observed. Four survivors showed a specific recurrent HL pattern in the low frequencies with normal hearing function or milder hearing loss in the higher frequencies (Figure 1). None of the survivors with deleterious HL were treated with carboplatin.

### 3.3. Laterality of Hearing Loss

In five survivors (5/8) with a Muenster ≥ 2b grade, HL occurred ipsilateral of local RMS treatment. In three survivors (3/8), HL occurred in both ears. These survivors had been treated for a medial orbital tumour with BT (as part of AMORE), a midline mandibular tumour with RT, and a nasopharyngeal RMS with RT.

### 3.4. Radiotherapy and Hearing Loss

Of the eight survivors diagnosed with deleterious HL (≥Muenster 2b), five survivors had been treated with RT, two with BT (as part of AMORE), and one with PT. Tumour localisation was in the mandibular (*n* = 2), nasopharynx (*n* = 2), and orbit (*n* = 1) for the RT-treated survivors, ear canal (*n* = 1) and orbit (*n* = 1) for the BT-treated survivors, and the infratemporal fossa for the survivor treated with PT. The cochlear irradiation dose was available for 23 of the 39 survivors treated at the Emma Children’s Hospital. We found a statistical difference between the maximum, mean, and minimum doses of RT at the cochleae of patients with and without HL (*p* = 0.015; *p* = 0.021 and *p* = 0.011, respectively, Figure 2). The maximum cochlear dose was most strongly correlated with HL occurrence (Pearson’s correlation 0.662, *p* < 0.01). HL (≥Muenster 2b) was significantly associated with the maximum cochlear irradiation dose (*p* = 0.047). In our series, HL (≥Muenster 2b) was especially observed when the maximum dose to the cochlea exceeded 19 Gy.

## 4. Discussion

The overall purpose of this study was to evaluate the hearing status and HL patterns of HNRMS survivors. Additionally, dose–effect relationships for the radiation dose to the cochlea and the development of HL were explored. For the first time, to our knowledge, a radiotherapy dose–effect relationship was explored in survivors of paediatric HNRMS. Although our study describes a small cohort, it includes a unique patient population that survived HNRMS with multimodality treatment including tertiary innovative approaches such as the brachytherapy (AMORE) protocol.

In the current study, we observed that up to 19% of HNRMS survivors had deleterious HL based on the Muenster criteria, 14% based on the CTCAE criteria, and 12% based on the SIOP criteria. However, it was challenging to grade audiograms, as ototoxicity grading scales are developed to monitor HL progression from high-to-low frequencies because most cases of paediatric oncology patients suffer from drug-induced (platinum-related) high-frequency hearing loss [37,38,39].

The HL prevalence in this study is in accordance with other studies. A previous study on survivors of HNRMS showed that clinically relevant hearing loss at speech frequencies occurred in 19% of survivors [14]. Schoot et al. (2016) [14] concluded that AMORE-based treatment resulted in less HL compared to RT; our study found that 2 of the 23 survivors treated with AMORE and 5 of the 16 RT-treated survivors developed deleterious HL (Muenster ≥ 2b). Lockney et al. (2016) [11] studied late toxicities in 30 HNRMS survivors (7.7 (1.2–14.4) years follow-up time) and showed that 20% of them had HL after a median dose of 50.4 (36–50.4) Gy delivered with intensity-modulated radiation therapy. In addition, Bass et al. (2016) [40] studied the effect of cranial irradiation in 235 children with brain tumours including craniopharyngioma, ependymoma, and juvenile pilocytic astrocytoma and found that during a median follow-up period of 9 years and a median of 11 post-RT audiograms per patient, 14% developed relevant HL.

Interestingly, in our cohort, four survivors had a peculiar but consistent low-frequency HL pattern with normal hearing function in the higher frequencies. Bass et al. (2016) [40] observed a so-called “tent-shaped” (loss in the low and high frequencies but normal in the mid-to-high-frequency range) HL pattern in two patients with brain tumours. Hence, it seems that this HL pattern is related to radiotherapy and/or local surgery, and not to chemotherapy or co-administered medication. No clear explanation for this pattern is available yet. It is conceivable that a local infiltrative growth of the tumour itself nearby the middle ear and/or cochlea as well as local surgery can destruct essential hearing structures. Another explanation might be that the outer and middle ear structures are exposed to radiation, potentially leading to chronic otitis externa/media, deep ulcerations, and problems in the external ear canal (stenosis and osteoradionecrosis), tympanic membrane perforation, and fibrosis [17,41]. Vascular insufficiency in the inner ear can develop weeks to months after irradiation, leading to the progressive degeneration and atrophy of sensory structures and even the fibrosis and ossification of the cochlear fluid space [15,22,42]. Addressing low-frequency HL is therefore highly important as this may influence speech and language developmental, educational, and cognitive outcomes in children, just as HL in the higher frequencies does [25].

Our study shows that HL (≥Muenster 2b) is significantly associated with the maximum cochlear irradiation dose (*p* = 0.047). In our series, HL (≥Muenster 2b) was especially observed when the maximum dose to the cochlea exceeded 19 Gy. This observation is confirmed by previous studies [23,40,43,44]. Recently, Keilty et al. (2021) [43] showed in patients with a brain tumour that the cumulative incidence of high-frequency HL (>4 kHz) was 50% or higher at 5 years after radiotherapy if the mean cochlear dose was >30 Gy. Additionally, Hua et al. (2008) [23] observed that the incidence of HL increased when the mean cochlear dose was greater than 40–45 Gy. Merchant et al. (2004) [19] reported that hearing loss in cranial irradiated children with primary brain tumours is uncommon in the first 4 years after cranial irradiation, although patients with shunts and supratentorial tumours appear to be at risk for low- and intermediate-frequency HL when the cochlear RT dose exceeds 32 Gy. To our knowledge, no literature exists in which the delineation of the cochlear was performed in BT (as part of AMORE) and RT treatment plans of HNRMS survivors. Based on the available literature and our findings, we recommend that one should try to minimise the radiation dose on the cochlea as much as possible to avoid hearing loss and keep the dose on the cochlea at least below 20 Gy if possible. Currently used radiotherapy techniques such as Volumetric-Modulated Arc Therapy (VMAT) [45,46], Cone-beam computed tomography [47], and improved radiotherapy masks [48] can reduce inaccuracies and prevent the unnecessary irradiation of organs at risk, for example, the cochlea.

Our data confirm that it is important to pursue audiological monitoring before but also long after head and neck irradiation, as HL can occur months to years after treatment. Awareness is needed to monitor this adverse effect during long-term follow-up [49] for a minimum of 10 years post-RT [40]. Age-appropriate audiological testing is recommended along with vertigo [50] and tinnitus screening [51]. Classifying audiological tests according to current ototoxicity grading classifications is challenging in this patient group, as these classifications are not developed for HL starting in the low and intermediate frequencies. Therefore, it is recommended to develop a classification system for radiation-induced hearing loss. We advise the assessment of such aberrant HL patterns by ear, nose, and throat (ENT) specialists or specialised audiologists since HL warrants evaluation and physical examination when patterns cannot be explained by administered therapies. The early detection of HL is important to determine if interventions for hearing function optimisation are necessary, such as hearing aids, remote microphone technologies, or cochlear implants to increase the quality of life of those children [52].

### Strengths and Limitations

Hearing function was studied in a relatively large Dutch HNRMS survivor cohort treated with different radiotherapy modalities. The survivors had a long follow-up time and were screened for late side effects on a regular basis. Their hearing status was evaluated by senior audiologists with experience in ototoxicity in the paediatric oncology field. For the first time, to our knowledge, cochleae were delineated in BT (as part of AMORE) and RT radiotherapy treatment plans, resulting in a more exact minimum, mean, and maximum cochlear radiotherapy dose.

Unfortunately, not all patients underwent audiological monitoring during follow-up but only patients complaining of HL or showing abnormalities at physical examination. Pre-treatment hearing evaluation was not performed in most patients due to lacking the recommendations for baseline testing in the treatment protocols. Hence, the presence of pre-existing hearing loss in a part of the patients cannot be excluded. In general, we recommend to always perform standardised audiological monitoring in patients with head and neck cancer that will undergo platinum treatment, radiotherapy, and/or ear, nose and throat surgery, including a baseline hearing assessment in future clinical practice. Dosimetry data were available for only a subset of the survivors (23/39, 59%) treated at the Emma Children’s Hospital. Furthermore, in contrast to chemotherapy data, data on ototoxic co-medication, like antibiotics and diuretics, were not available in all survivors. Although strong evidence is not available, we presume that the observed asymmetric HL is inflicted by local therapies. In the current study, the survivors were treated according to different treatment regimens. To our knowledge, the only potentially administered ototoxic chemotherapeutic agent in the survivors of our study could be carboplatin, but this had not been administered in the eight survivors with hearing loss (≥Muenster 2b). Furthermore, as shown in a study by Moke et al. (2021) [53], vincristine is a risk factor for HL in cisplatin-treated children. However, Riga et al. (2005) [54] and Lugassy et al. (1996) [55] found no evidence of the ototoxic effect of vincristine. To date, it is unknown whether the combination of cochlear irradiation and vincristine exposure has a detrimental effect on hearing. More research is needed in larger cohorts to study the effect of this treatment combination on HL. To our knowledge, no evidence exists for the relation between other chemotherapeutic agents received by the RMS survivors (ifosfamide, antracyclines, etoposide, vinorelbine, and/or cyclofosfamide) and hearing loss development.

Due to the limited number of patients with deleterious HL, it was not possible to perform multivariable analyses on dose–effect relationships. A larger sample size with complete cochlear irradiation data and audiological measurements during follow-up is necessary for a more accurate dose–effect analysis.

## 5. Conclusions

According to the Muenster grading, hearing loss occurred in up to 19% of HNRMS survivors at least 2 years had passed after treatment. Four survivors out of eight had peculiar low-frequency HL with normal/improved hearing function in higher frequencies. HL was significantly associated with the maximum cochlear irradiation dose. More research is needed on the dose–effect relationship of cochlear irradiation and the development of HL during long-term follow-up. Furthermore, the development of a classification scale to grade radiation-induced hearing loss is needed. Overall, audiological monitoring is recommended in HNRMS survivors before, during, and after radiotherapy to detect HL in an early state.

## Figures and Tables

**Figure 1 cancers-14-05749-f001:**
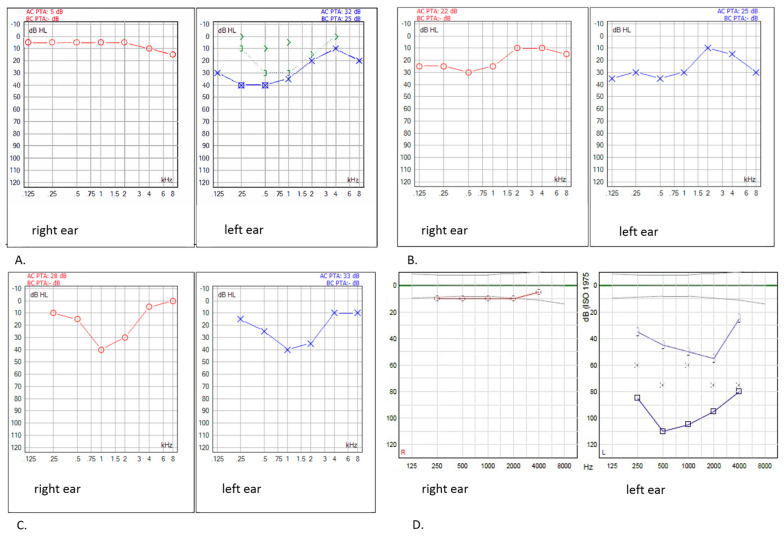
Patterns of hearing loss (HL) as shown per pure-tone audiogram in survivors with a left nasopharyngeal RMS (**A**), right nasopharyngeal RMS (**B**), right orbital RMS (**C**), and RMS in the left ear canal (**D**). Low-frequency HL with a mild HL/normal hearing function in higher frequencies is observed in survivor (**A**) left ear, survivors (**B**,**C**) both ears, and survivor (**D**) left ear.

**Figure 2 cancers-14-05749-f002:**
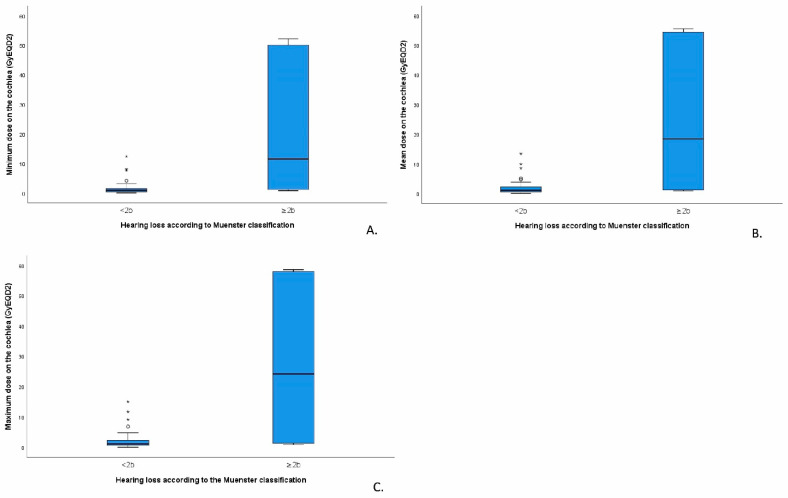
Boxplots showing minimum (**A**), mean (**B**) and maximum (**C**) irradiation dose on the cochlea (GyEQD2) in survivors without (Muenster < 2b) and with deleterious hearing loss (Muenster ≥ 2b). Outliers are indicated with *.

**Table 1 cancers-14-05749-t001:** Patient characteristics and audiologic outcomes.

Patient ID	RT Type	Sex	Diagnosis	Tumour Location	Tumour Side	Site	Age at Diagnosis (Years)	Time to FU (Years)	Treatment Protocol	Total Radiotherapy Dose (Gy)	Max. Cochlear Dose in Gy (Right)	Max. Cochlear Dose in Gy (Left)	Muenster Grade (Right)	Muenster Grade (Left)	SIOP Grade (Right)	SIOP Grade (Left)	CTCAE Grade (Right)	CTCAE Grade (Left)
1	AMORE	m	eRMS	Upper eyelid	right	NPM	6.0	13.4	MMT-95	40	0.7	0.4	1	1	0	0	0	0
2		f	eRMS	Ear canal	left	PM	10.1	11.5	RMS 2005	40	1.0	36.7	0	4	0	4	0	4
3		m	RMS ns	Parotid space	left	NPM	3.4	25.5	MT-89	46								
4		f	eRMS	Sinus maxillary, orbit, ethmoid	left	PM	2.4	17.3	MMT-95	50	1.02	1.38	0	0	0	0	0	0
5		f	eRMS	Masticator space	left	NPM	13.0	12.1	MMT-95	40	1.1	3.9	0		0		0	
6		m	eRMS	Pterygoid space	left	PM	2.1	22.4	CWS-91	45	11.2	1.9						
7		m	aRMS	Temporal	left	PM	1.8	11.4	RMS 2005	40	1.1	4.8	0	1	0	0	0	0
8		m	eRMS	Orbit	right	PM	7.7	11.6	MMT-95	45	1.3	0.7	0	0	0	0	0	1
9		f	eRMS	Nasal cavity	right	PM	1.3	21.0	MMT-95	45	1.1	1.0	1	1	0	0	0	0
10		f	eRMS	Orbit	left	orbit	8.0	8.3	RMS 2005	40	0.6	1.0						
11		m	eRMS	Orbit	right	orbit	5.5	6.6	RMS 2005	40	0.8	0.0						
12		m	eRMS/ botryoid	Nasal cavity	left	PM	3.2	2.0	RMS 2005	50	1.9	4.3	0	2a	0	1	0	1
13		f	aRMS	Orbit, ethmoidal sinus, sinus maxillary	left	PM	13.4	12.5	MMT-95	45	1.7	2.7	0	0	0	0	0	0
14		m	eRMS	Orbit	left	orbit	4.3	3.7	RMS 2005	40	0.4	0.5						
15		f	eRMS	Orbit	left	orbit	5.0	16.7	MMT-95	45	1.3	0.9	3a	3a	3	3	3	3
16		m	eRMS	Parotid	right	NPM	11.2	3.4	RMS 2005	40	11.7	1.0	2a	0	1	0	1	0
17		f	eRMS	Parotid space, mandibular	left	PM	5.8	25.3	MT-89	40			0	2a	0	1	0	1
18		m	aRMS	Nostril	left	NPM	7.5	15.5	MMT-95	40	0.4	0.5	0	0	0	0	0	0
19		m	eRMS	Orbit	right	orbit	10.2	2.9	RMS 2005	40			0	0	0	0	0	0
20		m	eRMS	Orbit	left	orbit	7.1	26.6	MMT-89	50	0.0	0.0	1	0	0	0	0	0
21		f	eRMS	Orbit, pterygopalatine fossa	right	PM	4.8	26.5	MMT-89	60			0	0	0	0	0	0
22		m	eRMS	Nasal cavity	left	PM	3.2	2.9	RMS 2005	68			1	2a	0	1	0	1
23		m	eRMS	Orbit	right	orbit	5.5	9.4	RMS 2005	40			0	0	0	0	0	0
24	Proton	f	eRMS	Orbit	right	orbit	6.3	9.0	RMS 2005	50.4			1	1	0	0	0	0
25		f	eRMS	Para-pharyngeal	right	PM	4.0	12.3	MMT-95	50			1	2a	0	1	0	1
26		m	eRMS	Mastoid /middle ear	left	PM	2.7	6.4	RMS 2005	55.8			0	1	0	0	0	0
27		f	aRMS	Infratemporal fossa	left	PM	4.6	11.2	RMS 2005				0	3c	0	3	0	3
28		f	eRMS	Maxilla	left	PM	4.0	3.6	RMS 2005	50.4			2a	0	1	0	1	0
29		f	eRMS	Orbit	left	orbit	2.0	7.3	RMS 2005				0	0	0	0	0	0
30		f	eRMS	Orbit	right	orbit	10.0	7.2	RMS 2005	45			0	0	0	0	0	0
31		f	eRMS	Orbit	right	orbit	8.2	8.7	RMS 2005	54			0	0	0	0	0	0
32		f	eRMS	Orbit	right	orbit	6.2	10.1	RMS 2005	50.4			0	0	0	0	0	0
33	Photon	m	eRMS	Para- pharyngeal	right	PM	3.3	26.4	MT-89	50.75			2a	0	1	0	1	0
34		m	eRMS	Angle of mandible	right	PM	0.04	13.5	MMT-95 RMS 2005	45			2b	0	1	0	2	0
35		f	eRMS	Nasopharynx	right	PM	4.0	12.2	RMS 2005	50.4	58.7	58.1	3a	3a	0	1	0	1
36		f	eRMS	Orbit	left	orbit	4.4	6.7	CWS-2007HR	50.4			0	0	0	0	0	0
37		m	eRMS	Nasopharynx, oropharynx	right/left	PM	4.5	6.7	RMS 2005	50.4	0.0	0.0	0	0	0	0	0	1
38		f	eRMS/ botryoid	Nasopharynx	right	PM	6.5	3.6	RMS 2005	50.4			0	0	0	0	0	0
39		m	eRMS	Orbit	right	orbit	7.2	3.4	RMS 2005	45	18.4	17.0						
40		m	eRMS	Sinus maxillary	left	PM	5.0	5.2	RMS 2005	50.4	9.1	15.0	0	0	0	0	0	0
41		f	aRMS	Mandible, ethmoid, Sella Turcica orbit	left	PM	4.4	16.2	MMT-95	54			3b	3c	4	4	3	4
42		m	eRMS	Cheek	right	NPM	0.5	5.8	RMS 2005	50.4	0.2	0.2	Not classifiable *					
43		f	eRMS	Orbit	left	orbit	7.2	9.9	RMS 2005	50.4	6.8	11.6	0	3a	0	3	0	3
44		m	eRMS	Sphenoid	NS	PM	5.1	13.5	MMT-95	50.4			0	1	0	0	0	0
45		f	eRMS/botryoid RMS	Nasopharynx	right	PM	6.5	6.0	RMS 2005	50.4			0	0	0	0	0	0
46		f	eRMS	Orbit	left	orbit	4.4	9.7	CWS-2007HR	50.4			0	0	0	0	0	0
47		m	eRMS	Orbit	right	orbit	4.9	14.4	MMT-95	45 + 40 brachy			0	0	0	0	0	0
48		f	eRMS	Orbit	left	Orbit	3.7	12.6	RMS 2005	45			1	0	0	0	0	0
49		m	eRMS	Nasopharynx	left	PM	13.1	9.8	RMS 2005 Avastin trial	55.8			0	3a	0	0	0	0

* Free field visual reinforcement audiometry: Hearing threshold at 0.5–4.0 kHz between 20–30 dB; Abbreviations: AMORE = Ablative surgery, MOuld technique brachytherapy and surgical REconstruction; aRMS = alveolar rhabdomyosarcoma; CTCAE = Common Terminology Criteria for Adverse Events; CWS = Cooperative Weichteilsarkom Study; RT = External Beam Photon RadioTherapy; eRMS = embryonal rhabdomyosarcoma; F = female; FU = follow-up; M = male; MMT = Malignant Mesenchymal Tumour; NPM = non-parameningeal; NS = not specified/unknown; PM = parameningeal; RMS = rhabdomyosarcoma; RMS 2005 = European Paediatric Soft Tissue Sarcoma Study Group RMS 2005; SIOP = Société Internationale d’Oncologie Pédiatrique; VRA = visual reinforcement audiometry.

**Table 2 cancers-14-05749-t002:** Clinical characteristics of patients with HNRMS.

	Total	Audiological Evaluation (*n* = 42)		Unknown/ Not Classifiable
		Muenster ≥ 2b HL	Muenster < 2b HL	
	(*n* = 49)	(*n* = 8)	(*n* = 34)	(*n* = 7)
Male (*n*)	24	2	17	5
Female	25	6	17	2
Age at diagnosis, median (range) (years)	5.0 (0.04–13.4)	4.8 (0.04–13.1)	5.1 (1.3–13.4)	4.3 (0.5–8.0)
Age at follow-up, median (range) (years)	16.3 (5.2–33.7)	18.9 (13.6–24.5)	16.6 (5.2–33.7)	12.1 (6.3–29.0)
Time to follow-up, median (ranges)	10.1 (2.0–26.6)	11.9 (9.8–16.7)	9.9 (2.0–26.6)	6.6 (3.4–25.5)
RMS histology subtype (*n*)				
Alveolar	5	2	3	0
Embryonal	43	6	31	6
Not specified	1	0	0	1
Tumour site (*n*)				
Parameningeal	25	6	18	1
Non-parameningeal	6	0	4	2
Orbital	18	2	12	4
Treatment protocol (*n*)				
MMT-95	13	3	10	0
MMT-89	5	0	4	1
CWS-91	1	0	0	1
CWS-2007HR	2	0	2	0
E*p*SSG RMS 2005	28	5	18	5
Radiotherapy subgroup (*n*)				
AMORE	23	2	16	5
RT	16	5	9	2
PT	9	1	8	0
Brachy + RT	1	0	1	0

Abbreviations: AMORE = Ablative surgery MOuld techniques after loading brachytherapy and surgical REconstruction; brachy = brachytherapy; HL = hearing loss; *n* = number; PT = proton radiotherapy; RT = photon radiotherapy.

## Data Availability

Data are available upon reasonable request.

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
