# Peer review of "Patterns of Hearing Loss in Irradiated Survivors of Head and Neck Rhabdomyosarcoma"

_cancers, 2022, doi:10.3390/cancers14235749_

Round 1
Reviewer 1 Report
this report is important and certainly of interest to the community of oncologists who treat children with radiation to the CNS
the report is limited by the sample size ( 49) It is also limited by the variability in time of follow up
none the less , few institutions will have access to a larger sample
Author Response
We would like to thank the reviewer for his/ her comments.
Please see the attachment for the point-by-point response to the reviewer's comments.

Reviewer 2 Report
Dear Editor,
I reviewed the article entitled “Hearing loss in survivors of head and neck rhabdomyosarcoma” by Diepstraten et al. discussing the long-term effect on hearing performance of treatments for head and neck tumors.
Despite the limits of the study, well stated by the authors, the paper has the merit to address a peculiar topic, in a systematic way, adding something to the current literature.
So, I consider this article suitable for publication in Cancers.
Author Response
We would like to thank the reviewer for his/ her positive feedback.
Please see the attachment.

Reviewer 3 Report
The study investigated an interesting topic on the effect of radiotherapy of HNRMS on hearing. Although the study is worthy, it is already well known that RT induces hearing loss. The results are not clearly presented and somewhat vague. The manuscript also needs English editing.
- The title is vague. It should indicate what did the authors investigate for the hearing loss in HNRMS. Pattern of hearing loss ? incidence ? the cause of hearing loss ?
- What is the cause of hearing loss in these patients ? Is it radiotherapy or combination with other chemotherapeutic agents ? It should be clearly stated what caused the hearing loss in survivors of HNRMS.
- The first paragraph in the Results section has no subheading.
- There is much limitation to draw a conclusion in this study. Every patients received different treatment. How about other causes of hearing loss such as chemotherapy agents ? Preop hearing is not checked. Were the children checked for tympanic membrane ? Multifactorial analysis is also needed for other causes that is related with hearing loss. The authors should make up these limitations. The conclusion that it is important to purse audiological monitoring is already well known
- Whole manuscript needs English editing. Please consider sourcing out the manuscript to a native English speaker.
Author Response
We would like to thank the reviewer for his/ her comments.
Please see the attachment for a point-by-point response to the reviewers comments.

Round 2
Reviewer 3 Report
The authors answered all the questions successfully. Well done by the authors. I recommend the authors to add the content of their response to question no.4 in the manuscript.
Author Response
We thank the reviewer for his/her valuable recommendation.
Please see the attachement for our point-by-point response.
